# Does SARS-CoV-2 Induce IgG4 Synthesis to Evade the Immune System?

**DOI:** 10.3390/biom13091338

**Published:** 2023-09-01

**Authors:** Alberto Rubio-Casillas, Elrashdy M. Redwan, Vladimir N. Uversky

**Affiliations:** 1Autlan Regional Hospital, Health Secretariat, Autlan 48900, Jalisco, Mexico; 2Biology Laboratory, Autlan Regional Preparatory School, University of Guadalajara, Autlan 48900, Jalisco, Mexico; 3Biological Science Department, Faculty of Science, King Abdulaziz University, P.O. Box 80203, Jeddah 21589, Saudi Arabia; lradwan@kau.edu.sa; 4Therapeutic and Protective Proteins Laboratory, Protein Research Department, Genetic Engineering and Biotechnology Research Institute, City for Scientific Research and Technology Applications, New Borg El-Arab 21934, Alexandria, Egypt; 5Department of Molecular Medicine, Morsani College of Medicine, University of South Florida, Tampa, FL 33612, USA; 6USF Health Byrd Alzheimer’s Research Institute, Morsani College of Medicine, University of South Florida, Tampa, FL 33612, USA

**Keywords:** SARS-CoV-2, IgG4, Tregs, long COVID, immune tolerance

## Abstract

SARS-CoV-2, the virus that causes the COVID-19 disease, has been shown to cause immune suppression in certain individuals. This can manifest as a reduced ability of the host’s immune system to effectively control the infection. Studies have reported that patients with COVID-19 can exhibit a decline in white blood cell counts, including natural killer cells and T cells, which are integral components of the immune system’s response to viral pathogens. These cells play critical roles in the immune response to viral infections, and their depletion can make it harder for the body to mount an effective defense against the virus. Additionally, the virus can also directly infect immune cells, further compromising their ability to function. Some individuals with severe COVID-19 pneumonia may develop a “cytokine storm”, an overactive immune response that may result in tissue damage and organ malfunction. The underlying mechanisms of immune suppression in SARS-CoV-2 are not entirely understood at this time, and research is being conducted to gain a more comprehensive understanding. Research has shown that severe SARS-CoV-2 infection promotes the synthesis of IgG4 antibodies. In this study, we propose the hypothesis that IgG4 antibodies produced by B cells in response to infection by SARS-CoV-2 generate immunological tolerance, which prevents its elimination and leads to persistent and chronic infection. In summary, we believe that this constitutes another immune evasion mechanism that bears striking similarities to that developed by cancer cells to evade immune surveillance.

## 1. Introduction

Viruses are extremely complex molecular entities that can go through intra-host evolution to evolve tactics that successfully subvert the host immune system and sustain chronic infection by constantly reproducing, creating latent reservoirs, or merging into the genome of the host cell. The virus’s persistent immune stimulation and/or pathologic effects last until the infection resolves or the host is killed—if the immune system is unable to eliminate the virus [1]. Some viruses can persist within a host for long periods by establishing a balance with the host’s immune response. This is known as a “metastable virus-host immune response interaction equilibrium”. Examples of such viruses include the hepatitis B virus (HBV), the human immunodeficiency virus (HIV), and the hepatitis C virus (HCV). These viruses have the ability to evade the host’s antiviral immunity by remaining in a latent state within host cells, enabling them to avoid identification and destruction by the immune system of the host. This allows the virus to persist within the host for prolonged periods, making it difficult to eliminate the infection [2].

In support of the possibility that SARS-CoV-2 can also establish this persistence within the host, it was recently reported that a small number of COVID-19 patients appeared to heal from acute disease, although their condition still progressed and they eventually died [3]. Autopsies were conducted on 27 patients who appeared to have recovered from COVID-19 but whose clinical conditions had gotten progressively worse, regardless of nasopharyngeal swab viral negativity or PCR tests. Notwithstanding the notorious virological recovery, pathological changes in the lungs were comparable to those detected in acute COVID-19 patients. Forensic analysis of the lungs indicated that pneumonia was the main cause of death. This research demonstrated that these individuals still had cells infected with SARS-CoV-2 in their lungs, especially in the para-bronchial glands and bronchial cartilage, notwithstanding the negative test results. This substantiates the hypothesis that such patients did not completely recover from the infection. The lack of SARS-CoV-2 in the respiratory epithelium may explain why these patients had negative PCR tests when their bronchoalveolar lavage samples were analyzed [3]. It is also possible that the observed symptoms were caused not by the virion itself but by the viral S and/or N proteins, which stimulated some subtype of macrophages that released pathological mediators and cytokines.

## 2. IgG4 Antibodies Induced by SARS-CoV-2 May Help It Evade the Immune System

Human Immunoglobulin G (IgG) consists of four subcategories (IgGl, IgG2, IgG3, and IgG4), which, although highly conserved, differ in their constant regions, particularly in their hinges and upper CH2 domains, and are distinguished by the immunogenicity of their heavy chains [4,5,6]. It has been demonstrated that intrinsic characteristics of the antibody immune response, including antigen specificity, antigen persistence, antibody glycosylation, antibody affinity, antibody avidity, neutralizing antibody, and immunoglobulin (Ig) G subclass, i.e., specific characteristics that primarily determine the antibody durability and neutralization capacity, affect the course of SARS-CoV-2 infection and COVID-19 prognosis [7,8,9,10]. A poor COVID-19 clinical evolution partly relies on an unbalanced antibody response, resulting in increased systemic inflammation due to defective viral neutralization [8]. A review of the major characteristics of this unusual antibody was recently published [11].

There are few publications on IgG4 production following SARS-CoV-2 infection. The IgG1 and IgG3 subclasses are the most prevalent in COVID-19 patients [10,12,13]. Since IgG4 is not expected to have a role in this viral infection, many studies, particularly those published in 2019–2022, did not include it during estimation. Omicron breakthrough infections were reported to increase IgG4 production by up to three times in mRNA-vaccinated individuals, showing that SARS-CoV-2 infection can also stimulate IgG4 synthesis [14].

A recent study sought to identify a connection between COVID-19-associated mortality and IgG subclasses, given their various inflammatory qualities. A hospital in Italy received 138 patients with COVID-19 (41 females, 31.3%) from June to December 2020. Daily data collection was used to continuously follow up on these patients. IgG1, IgG2, IgG3, and IgG4 subtypes increased at baseline in 8 (6%), 6, (5%), 6 (5%), and 13 (10%) individuals, respectively. A total of 30 patients (23% of the total) passed away after 30 days [15]. Interleukin (IL)-6, C-reactive protein (CRP), and serum IgG4 levels were significantly higher in non-survivors. Notably, an IgG4/IgG1 ratio > 0.05 and a serum IgG4 concentration > 700 mg/dL were linked to a considerably higher 30-day death rate, and serum IgG4 concentration was significantly associated with levels of IL-6 [15], a recognized forecaster of COVID-19-linked mortality [16,17,18] whose synthesis is induced by the open reading frame protein 8 (ORF8) of SARS-CoV-2 [19]. IL-6 also increases the serum levels of IgG4 and other IgG subtypes [20,21]. However, we should also mention here that although Della-Torre et al. reported that serum IgG4 levels can potentially predict COVID-19-related mortality [15], it was also indicated [22] that this conclusion may be far-fetched due to the small number of enrolled patients (128 participants) and the unique properties of IgG4 protein, whose levels may vary significantly in healthy individuals [23].

Based on the research that is currently available, IgG4 antibodies can aggravate COVID-19 cases via at least two pathways [15]. One hypothesis is that individuals with high IgG4 levels may be more susceptible to infection with SARS-CoV-2 because anti-spike IgG4 has demonstrated weak in vitro neutralizing ability compared with IgG1, IgG2, and IgG3 antibodies [14]. On the other hand, anti-IFNγ auto-antibodies were identified as an IgG4 subclass in people with adult-onset immunodeficiencies similar to advanced HIV infection. Researchers detected a disproportionately higher concentration of anti-IFNγ auto-antibodies in 88% of Asian adults with numerous opportunistic infections [24]. It is therefore probable that the anti-IFN antibodies linked to decreased anti-SARS-CoV-2 immunity and potentially fatal COVID-19 pneumonia may be an IgG4 subclass [15]. Since the start of the COVID-19 pandemic, several studies have identified a relationship between pre-existing autoantibodies against type I interferons (IFNs) and the severity of COVID-19 [7,25,26,27]. These autoantibodies can inhibit the effectiveness of the corresponding type I IFNs to prevent SARS-CoV-2 infections in vitro [7] and in vivo. Although SARS-CoV-2 infection induces the synthesis of IgG4, in about 10% of severe cases of COVID-19 pneumonia, the patients had high pre-existing levels of neutralizing auto-antibodies against type I IFNs, which were responsible for the severity of the disease [7].

Importantly, it has been demonstrated that interferon gamma (IFNγ) significantly inhibits IL-6-induced antibody secretion [20]. This evidence suggests that in non-complicated COVID-19 cases, IFNγ can halt excessive IL-6 production. However, in severe cases, pre-existing auto-anti-IFNγ antibodies contribute to a severe clinical course because these auto-antibodies block normal INFγ function. Under such conditions, SARS-CoV-2-induced IL-6 release increases uncontrollably, thus contributing to the development of the cytokine storm associated with severe cases of COVID-19. This is one of the many strategies for the induction and propagation of the cytokine storm.

A prospective study demonstrated that blood IgG4 levels reflect poor COVID-19 prognosis. That research identified IgG4 antibodies as a potential additional underappreciated characteristic of antibody responses against SARS-CoV-2 that is associated with COVID-19 prognosis [15]. Another study found that patients with COVID-19 who passed away between 8 and 14 days or between 15 and 21 days also had higher anti-RBD (receptor binding domain) IgG4 levels compared with those who recovered (*p*  <  0.05), proposing that some individuals who are in a life-threatening condition can trigger an IgG4 to RBD antibody response in the first weeks after the appearance of symptoms [12]. In that investigation, more than half of the blood samples from patients who died tested positive for IgG4 antibodies; in contrast, the majority of patients who recovered from COVID-19 tested negative for IgG4 over the same period [12].

Principal antibody responses to many viral illnesses are mainly represented by the IgG1 and IgG3 subclasses [28,29]. More recently, these subclasses have also been linked to SARS-CoV-2 infections [30,31]. However, it was pointed out that IgG4 generally represents a minor component of the total immunoglobulin response and is mostly induced in response to continuous antigenic stimulation [32]. In fact, in viral respiratory illnesses, IgG1 and IgG3 responses are associated with immunological processes such as viral neutralization, opsonization, and complement activation [28]. One research group examined the SARS-CoV-2 RBD IgG isotypes in sera from non-critical COVID-19 patients. As predicted, strong IgG1 and IgG3 antibody responses specific to RBD predominated, in contrast to weaker IgG4 responses [12]. Similar to these results, patients with non-severe COVID-19 evaluated in the USA generated RBD-specific IgG1 and IgG3 antibodies early during acute infection, with no discernible IgG2 or IgG4 production [30]. Comparable antibody reactions with significant IgG1 and IgG3 reactivity in sera from individuals who tested positive for SARS-CoV-2 infection were also documented in Italy [31]. On the other hand, SARS-CoV-2 infections induce IgG4 production, and high serum IgG4 concentrations (>700 mg/dL) were linked to a considerably higher 30-day death rate and were significantly correlated with IL-6 levels [15], a known forecaster of COVID-19-related mortality [16,17,18]. We suggest that IgG4 antibodies induced by infection with SARS-CoV-2 exacerbate a pre-existing condition with elevated levels of IgG4 autoantibodies in some individuals, thus complicating the clinical outcome and resulting in a higher mortality rate.

## 3. Mechanisms of IgG4-Induced Immune Evasion in SARS-CoV-2 Infection

A regulated immune response that recognizes and destroys the invading pathogen while reducing collateral tissue damage that might arise from an overactive immune response is necessary for the host to survive an infection. However, for the majority of pathogens to successfully transmit, they must avoid being destroyed by the host immune system. Therefore, immunological modulation during an infection may develop as a result of the host’s reaction to the infectious process in an effort to preserve or restore a homeostatic environment, or it may be deliberately generated by the pathogen to increase the chances of its survival. Many viruses have developed different ways to take advantage of the regulatory system of the host, creating conditions that ensure their survival for long periods [33]. In the case of SARS-CoV-2, these tactics involve antigenic variation to evade humoral and cellular immunity, interference with antigen processing or presentation, and subversion of phagocytosis or death by innate immune system cells (reviewed in [34]).

In this study, we propose the hypothesis that SARS-CoV-2 induces IgG4 synthesis to promote immune tolerance, thus evading immune surveillance and permitting unopposed viral replication. To gain insights into how IgG4 can be associated with a virus-induced immune suppression mechanism, it is necessary to understand how this antibody is produced, identify the cells that produce it, and determine the other immune mechanisms involved. IgG4 is produced by B cells in response to the release of high concentrations of interleukin 10 (IL-10) by regulatory T cells [35]. Tregs are necessary for preserving immunologic homeostasis, self-tolerance, and stopping uncontrolled immunological reactions [36]. Through a variety of effector pathways, Tregs control the activity of various innate and adaptive immune system branches [37]. Additionally, specific populations of “tissue Tregs” regulate homeostasis in a variety of non-immunological tissues, reducing inflammation and fostering proper tissue regeneration [38]. However, Tregs can be harmful, as indicated by their suppression of efficient cytotoxic responses in cancer, where they assume a unique phenotype [39,40,41]. They can also generate contradictory responses to antiviral infections [42,43].

To prolong their life, viruses frequently induce regulatory responses that are typically linked to the inhibition of the host’s effector immune responses. This can be accomplished directly by stimulating the production of host immune regulatory cytokines such as IL-10 and transforming growth factor-(TGF) by innate immune cells in response to molecules derived from the pathogens, or indirectly by stimulating the development of regulatory cells (Tregs) [33].

Increased numbers of Tregs were found in patients with severe COVID-19 who were admitted to the intensive care unit (ICU). These cells expressed high levels of the transcription factor FoxP3, which is associated with negative outcomes. Aberrant Tregs contribute to COVID-19 physiopathology, as patients with fewer Tregs and FoxP3 have better clinical outcomes [44]. Interestingly, abnormal Tregs have a startling resemblance to Tregs that infiltrate tumors and suppress anticancer responses. These findings imply that Tregs may have negative effects on COVID-19 by inhibiting antiviral T-cell responses during the disease’s acute phase and by directly promoting inflammation. The increase in Tregs is induced by IL-6 [44], a cytokine induced by the ORF8 protein of SARS-CoV-2 [19]. IL-6 is also necessary for increasing the immune suppression ability of the RORγ^+^ Treg subpopulation [45]. Interestingly, IL-6 also increases IgG4 serum levels [20,21].

In summary, our hypothesis postulates that during acute infection, SARS-CoV-2 induces Treg activation and proliferation through excessive IL-6 signaling. Tregs then release IL-10, which induces B cells to produce IgG4 antibodies to promote tolerance and favor pathogen survival by blocking antiviral responses. The proposed mechanism is shown in Figure 1.

We also suggest that, through this IgG4-mediated tolerance mechanism, SARS-CoV-2 has developed another immune evasion mechanism that is very similar to that of cancer cells by promoting RORγ^+^ Treg-mediated immune tolerance. In a previous study [44], researchers focused their investigation on aspects of Tregs that may be similar in both malignancies and SARS-CoV-2 infections. One such component was hypoxia, a characteristic of cancers and a significant contributor to severe COVID-19 [46] that can enhance Treg suppressive activity [47]. Accordingly, significant levels of lactic acid are found in COVID-19 patients [48] and in tumors, where it has been shown to impact Treg function [49]. Finally, since all samples from severely affected patients profiled were obtained during the cytokine storm phase, it is possible that COVID-19 Tregs excessively inhibited the antiviral response, leading to a secondary re-expansion of the virus. This possibility is supported by the overexpression of FoxP3 and Treg effector molecules, as well as by their resemblance to suppressive tumor Tregs [44].

## 4. Conclusions

To combat infections, the immune system has developed a variety of innate and adaptive strategies. Similarly, to guarantee their permanence within the host, viruses have developed sophisticated mechanisms to elude the immune system, generating chronic infections that are rarely eliminated [50]. We propose that when SARS-CoV-2 infects the respiratory system, the viral ORF8 protein induces IL-6 production, thus altering the B lymphocytes’ normal phenotype and function. These cells then synthesize high amounts of IgG4 antibodies. This antibody has weak in vitro neutralizing potential compared with IgG1, IgG2, and IgG3 antibodies [14,29,30]. Recent studies confirmed that a class change from IgG3 to IgG4 was linked to a decreased ability of the spike-specific antibodies to trigger complement deposition and antibody-dependent cellular phagocytosis [14]. The proposed mechanism through which the IgG4 antibody blocks IgG3 attachment to its Fc receptor, and thus inhibits viral phagocytosis, is shown in Figure 2. This model is supported by the known capability of human IgG4 to interact with the Fc parts of all IgG subclasses [51].

By using this IgG4-mediated tolerance mechanism, SARS-CoV-2 avoids being detected and attacked by the immune system. In other words, our immune system is hijacked and forced to “tolerate” or “ignore” the virus, thus possibly allowing chronic infection. The IgG4-induced tolerance can produce a poor immune response against SARS-CoV-2 when patients suffer re-infection. The virus can also infect cells for a long time, causing chronic infection. It is important to note that SARS-CoV-2 employs other additional evasion mechanisms to avoid immune surveillance and attack, including interferon synthesis inhibition [52,53,54,55,56], antigen presentation disruption [57,58], antibody evasion via nanotube construction [59,60], and induced lymphopenia via syncytia generation [61,62,63].

Finally, the probable IgG4-mediated evasion mechanism induced by SARS-CoV2 is remarkably similar to that evolved by cancer cells to avoid immune surveillance and attack. Researchers have investigated malignant melanoma and discovered that tumor-specific IgG4 is produced locally in tumor tissues. Additionally, they discovered that, unlike cancer-specific IgG1, cancer-specific IgG4 does not activate two immunological mechanisms that use antibodies to recognize and eliminate cancer cells [64]. Furthermore, IgG1 antibodies prevented the spread of cancer in an in vivo model, but IgG4 did not. While IgG1 antibodies are responsible for mediating tumor cell death, IgG4 antibodies can obstruct this process. These results highlight an aspect of tumor-driven immunological escape that has not previously been studied: IgG4 production driven by tumors restricts effector immune cell activity against them [64]. IgG4 antibodies are significant and are required for cancer immune evasion [65]. B cells with high IgG4 concentrations were found in both serum samples and malignant cells in a cohort of patients with esophageal cancer. Additionally, both increased cancer malignancy and poor prognosis were highly correlated with higher IgG4 levels, which appear to be linked to more aggressive cancer growth. It was also found that IgG4 can compete with IgG1 for in vitro attachment to Fc receptors found in some immune cells (as depicted in Figure 3). The normal immune responses against cancer cells, such as cell and complement cytotoxicity and cell phagocytosis that are mediated by IgG1 antibodies, are inhibited as a result of this competition [65].

From our hypothesis, it can be noted that IL-6 represents an essential cytokine that promotes a cascade of events favoring SARS-CoV-2 pathogenesis, as interleukin-6 (IL-6) serum levels were linked to the severity of clinical symptoms and poor COVID-19 prognosis [15,66,67]. It was hypothesized that by inhibiting the IL-6 signaling pathway, a better clinical outcome could be achieved. One study assessed the safety and efficacy of IL-6 blockade with sarilumab in patients with severe COVID-19 pneumonia and systemic hyperinflammation. A total of 28 patients received sarilumab treatment, whereas another 28 patients, who received only routine care, served as controls. By day 28 of follow-up, 61% of patients who received sarilumab had improved clinically, while 7% had passed away. These findings (64% clinical improvement, 18% mortality; p = NS) did not differ significantly from the comparison group. Clinical improvement in patients receiving sarilumab was indicated by baseline PaO_2_/FiO_2_ ratio > 100 mm Hg and lung consolidation < 17% at Computed Tomography (CT) examination [16]. It was proposed that the late administration of the IL-6 blockers represented a major factor in treatment failure. A subsequent study by these researchers aimed at identifying a “window of therapeutic opportunity” for enhancing IL-6 blockage effectiveness in COVID-19. The study included 107 individuals receiving IL-6 inhibitors and 103 patients receiving standard care. When started in patients with 100 mmHg of PaO_2_/FiO_2_, treatment with IL-6 inhibitors was linked to a significantly higher survival rate compared with conventional treatment after an average of 106 days of monitoring (range 3–186; *p* < 0.001). The researchers came to the conclusion that IL-6-blocking treatments have a greater likelihood of improving survival in COVID-19 patients who are hyper-inflamed when started before the onset of acute respiratory failure [18].

IL-6 emerges as a key cytokine in the multiplication and immune evasion of both SARS-CoV-2 and cancer cells. Just to cite an example: one study found that IL-6 efficiently protects gastric cancer cells from apoptosis caused by hydrogen peroxide (H_2_O_2_) by increasing the production of Mcl-1 (an anti-apoptotic protein) [68]. The similarity between our proposed mechanism for IgG4-mediated immune evasion by SARS-CoV-2 and that evolved by cancer cells suggests that viruses are more complex than usually assumed. It is not known whether viruses or cancer cells were the first to develop this ingenious evasion mechanism, but we are convinced that studying the molecular mechanisms that govern these processes can have therapeutic implications in the treatment of both viral infections and cancers.

## Figures and Tables

**Figure 1 biomolecules-13-01338-f001:**
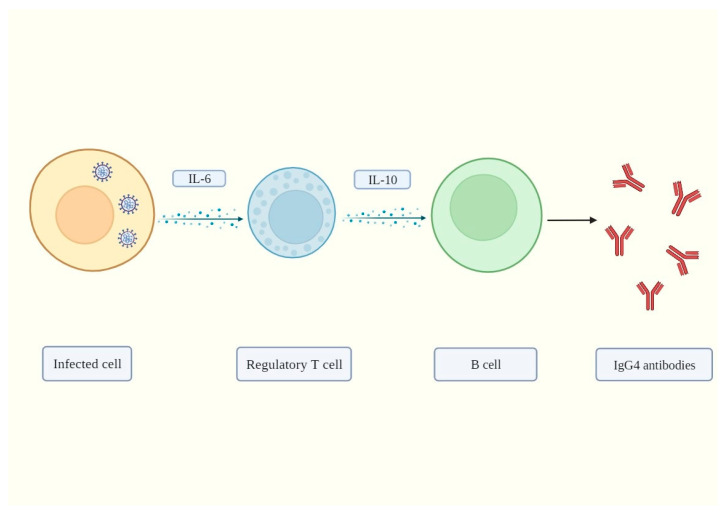
SARS-CoV-2 infects a permissive cell, and its ORF-8 protein induces interleukin 6 (IL-6) synthesis. This cytokine then stimulates the activation and excessive proliferation of regulatory T cells (Tregs), which in turn release interleukin 10 (IL-10). In adequate concentrations, IL-10 exerts anti-inflammatory functions. However, in higher concentrations, it exerts pro-inflammatory effects. IL-10 then induces IgG4 antibody production. These antibodies can block normal antiviral responses, such as complement-dependent cytotoxicity (CDC), antibody-dependent cell-mediated cytotoxicity (ADCC), and antibody-dependent cell phagocytosis (ADCP), thus favoring viral replication and persistence in the host. Created with BioRender.com.

**Figure 2 biomolecules-13-01338-f002:**
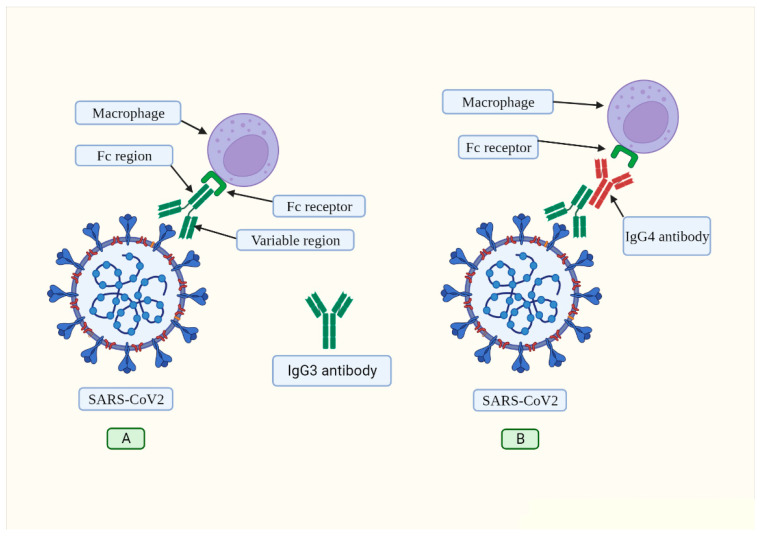
(**A**) Under normal conditions, the IgG3 antibody binds to the spike protein via its variable region. This antibody has a constant region (Fc) that is recognized by the corresponding receptor found on macrophages and other immune cells. This mechanism is called “opsonization” and marks foreign pathogens for phagocyte destruction. (**B**) SARS-CoV-2 induces IL-6 production, altering the normal phenotype of B cells, and making them produce IgG4 antibodies (depicted in red). The constant region of the IgG4 antibody binds to the constant region of IgG3, thus preventing the binding of said antibody to its receptor located on the macrophage. In this way, IgG3 effector functions are blocked. Created with BioRender.com.

**Figure 3 biomolecules-13-01338-f003:**
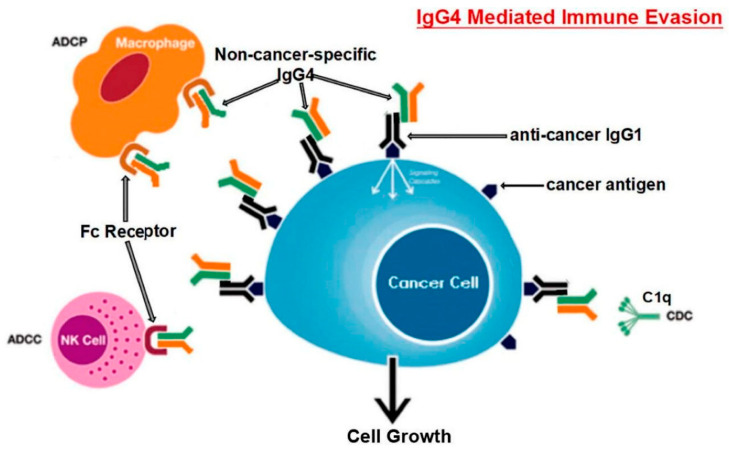
Schematic representation of the hypothesized mechanism for immune evasion by cancer cells using B lymphocyte-produced IgG4. Long-term contact with tumor antigens induces B cells to switch classes and produce IgG4. Due to its unique structural and biological characteristics, increased IgG4 in the cancer microenvironment creates an effective immune evasion mechanism for the disease. The terms antibody-dependent cell-mediated cytotoxicity (ADCC), antibody-dependent cell phagocytosis (ADCP), complement-dependent cytotoxicity (CDC), and natural killer cells (NK) are abbreviations of these processes. Adapted from [65]. This article is open access and is distributed under the Creative Commons Attribution Non-Commercial (CC BY-NC 4.0) license, which enables others to distribute, remix, adapt, and build upon it for non-commercial purposes and to license their derivative works under different conditions as long as the original work is properly cited, due credit, any changes are noted, and the use is for non-commercial purposes.

## Data Availability

No new data were generated or analyzed in this study.

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
