# Peer review of "Does SARS-CoV-2 Induce IgG4 Synthesis to Evade the Immune System?"

_biomolecules, 2023, doi:10.3390/biom13091338_

Round 1

Reviewer 1 Report

The authors present the hypothesis that SARS-CoV-2 infection can induce the production of IgG4 antibodies which contribute to an immunosuppressive state. The hypothesis is supported by data showing that increased severity of disease and mortality from SARS-CoV-2 infection is often correlated with an increase in IgG4 levels. Mechanistically, the authors propose that the increase in IgG4 production is induced by cytokines secreted by prevalent Tregs and Bregs. IgG4 has been demonstrated to have poorer neutralization activity compared to other subtypes and the ability to interact with Fcs of other IgG subclasses, inhibiting the latter’s effector functions and contributing to viral immune evasion.

Overall, the hypothesis presented is supported by previous research done on SARS-CoV-2 and parallels are seen in other diseases, most notably cancer. Further investigation of this hypothesis could contribute to our understanding of the immune response to SARS-CoV-2 and the development of new therapeutics.

Some comments and concerns include the following.

“IgG4 is known to be produced by a variety of immune cells, including Th2 cells, Tregs, and Bregs when the cytokine IL-10 levels are elevated” (lines 167-168) is incorrect and/or misleading. Presumably, the authors mean that these cell types induce the production of IgG4, but T cells cannot produce Abs directly. Additionally, the rest of this section discusses Tregs without a clear connection to IgG4-induced immune evasion. The mechanistic role of IgG4s in this process is only discussed later in the conclusion.

While the hypothesis presented is plausible, numerous other hypotheses explaining the correlation between increased severity of disease and mortality from SARS-CoV-2 infection and an increase in IgG4 levels are also possible. In just one example of such an alternative hypothesis, an increase in IgG4 levels, relative to IgG1 and IgG3, may lead to higher occupancy of antigenic binding sites by IgG4 and a subsequent overall reduction in antibody-mediated effectors, since IgG4 binds with lower affinity to activating Fc receptors and triggers relatively less potent effector functions. Such a mechanism could occur independent of or coincidentally to the authors’ hypothesis. Regardless, the authors should consider presenting some alternative hypotheses and an explanation of why they prefer the one they have presented here.

Editing required prior to publication.

Author Response

Reviewer 1

“IgG4 is known to be produced by a variety of immune cells, including Th2 cells, Tregs, and Bregs when the cytokine IL-10 levels are elevated” (lines 167-168) is incorrect and/or misleading. Presumably, the authors mean that these cell types induce the production of IgG4, but T cells cannot produce Abs directly. 

R: We deeply appreciate Reviewer 1 for this important observation. We have corrected this text on page 4 as follows:

IgG4 is known to be produced by B cells in response to high interleukin 10 (IL-10) concentrations released by regulatory T cells [35].

“Additionally, the rest of this section discusses Tregs without a clear connection to IgG4-induced immune evasion.”

R: We thank to reviewer 1 for this important clarification. We have explained our hypothesis in a clearer way and added a new figure to illustrate it. In page 5 we wrote:

In summary, our hypothesis postulates that during the acute infection, the SARS-CoV-2 induces Tregs activation and proliferation though excessive IL-6 signaling. Tregs then release IL-10 that induces B cells to produce IgG4 antibodies to promote tolerance and favor pathogen survival by blocking antiviral responses. The proposed mechanism is shown in Figure 1.

Figure 1. SARS-CoV-2 infects a permissive cell and its ORF-8 protein induces interleukin 6 (IL-6) synthesis. This cytokine then stimulated the activation and excessive proliferation of regulatory T cells (Tregs), which in turn release interleukin 10 (IL-10). In adequate concentrations, IL-10 exerts anti-inflammatory functions. However, in higher concentrations, it exerts pro-inflammatory effects. The IgG4 antibodies could block normal antiviral responses, such as antibody-dependent cell-mediated cytotoxicity (ADCC) and antibody-dependent cell phagocytosis (ADCP), thus favoring viral replication and persistence in the host, leading to chronic infection.

“ While the hypothesis presented is plausible, numerous other hypotheses explaining the correlation between increased severity of disease and mortality from SARS-CoV-2 infection and an increase in IgG4 levels are also possible. In just one example of such an alternative hypothesis, an increase in IgG4 levels, relative to IgG1 and IgG3, may lead to higher occupancy of antigenic binding sites by IgG4 and a subsequent overall reduction in antibody-mediated effectors, since IgG4 binds with lower affinity to activating Fc receptors and triggers relatively less potent effector functions. Such a mechanism could occur independent of or coincidentally to the authors’ hypothesis. Regardless, the authors should consider presenting some alternative hypotheses and an explanation of why they prefer the one they have presented here.”

R: We apologize because our hypothesis was not sufficiently explained. We have included the new figure 1 for better explanation. We believe that our schematic representation shows the main events and includes several of the mechanisms that were apparently unrelated. It highlights the fundamental role of IL-6 in the activation of a cascade of events that influence the pathogenesis of COVID-19 and the immune evasion by SARS-CoV-2.

Reviewer 2 Report

In this manuscript, Rubio-Casillas et al. described a hypothesis that SARS-CoV-2 induce IgG4 synthesis to evade the immune system, which is an interesting try to give new explanation for SARS-2 induced long infection as well as immune evading.

A few of suggestion for this manuscript:

A bit longer introduction for IgG4 antibody would be helpful for the common readers. It should pay more attention on the differences between IgG4 and IgG1, 2,3.

Actually, the SARS-2 must use multiple mechanisms to evade the immune attack or surveillance, the IgG4 possibly is part of the mechanism, so the authors should state clearly that it is only part of the mechanism.

A table of “IgG4 and virus-induced immune evading” could be helpful to summarize the field.

The last part, the authors compared the similarity between SARS-2 and cancer -induced IgG4, which the reviewer think it would be great to list some suggestion, how should we take advantage for this mechanism to treat that kind of diseases.

Author Response

“ A bit longer introduction for IgG4 antibody would be helpful for the common readers. It should pay more attention on the differences between IgG4 and IgG1, 2,3.

R: We thank reviewer #2 for this suggestion. However, there are other publications that have recently addressed this issue. Therefore, in page 2 we wrote:

A review of the main characteristics of this unusual antibody was recently published [[11].

  • Uversky, V.N.; Redwan, E.M.; Makis, W.; Rubio-Casillas, A. IgG4 Antibodies Induced by Repeated Vaccination May Generate Immune Tolerance to the SARS-CoV-2 Spike Protein. Vaccines 2023, 11, 991.

“Actually, the SARS-2 must use multiple mechanisms to evade the immune attack or surveillance, the IgG4 possibly is part of the mechanism, so the authors should state clearly that it is only part of the mechanism”.

R: We thank to reviewer for this suggestion. In page 7 we wrote:

It is important to note that, in addition to this, SARS-CoV-2 employs a number of evasion mechanisms to avoid immune surveillance and attack, including interferon synthesis inhibition [52][53][54][55][56], antigen presentation disruption [57][58], antibody evasion via nanotube construction [59][60], and induced lymphopenia via syncytia generation [61][62][63].

    “The last part, the authors compared the similarity between SARS-2 and cancer -induced IgG4, which the reviewer think it would be great to list some suggestion, how should we take advantage for this mechanism to treat that kind of diseases”.

R: We sincerely thank to reviewer # 2 for such important comment. We have added this text in the final part of the Conclusions.

From our hypothesis, it can be noted that IL-6 is an essential cytokine that promotes a cascade of events that favor SARS-CoV-2 pathogenesis, as interleukin-6 (IL-6) serum levels were linked to COVID-19 patients' severe clinical symptoms and poor prognosis [15][66][67]. It was hypothesized that by inhibiting the IL-6 signaling pathway, a better clinical outcome could be achieved. One study assessed the safety and efficacy of interleukin (IL)−6 blockade with sarilumab in patients with severe COVID-19 pneumonia and systemic hyperinflammation. 28 patients received sarilumab treatment, whereas 28 current patients who received only routine care served as controls. At day 28 of follow-up, 61% of patients who received sarilumab had improved clinically, while 7% had passed away. These findings (clinical improvement 64%, mortality 18%; p=NS) did not differ significantly from the comparison group. Clinical improvement in patients receiving sarilumab was indicated by baseline PaO2/FiO2 ratio >100 mm Hg and lung consolidation <17% at Computed Tomography (CT) examination [16]. It is evident that late IL-6 blockers administration is a major factor in treatment failure. A subsequent work by these researchers aimed to identify " window of therapeutic opportunity" for enhancing IL-6 blocking effectiveness in COVID-19. The study included 107 individuals receiving IL-6 inhibitors and 103 patients receiving standard of care. When started in patients with a PaO2/FiO2 100 mmHg, treatment with IL-6 inhibitors was linked to a significantly higher survival rate than conventional treatment after an average of 106 days of monitoring (range 3-186; p 0.001). The researchers came to the conclusion that IL-6 blocking treatments have a greater likelihood to improve survival in COVID-19 patients who are hyper-inflamed when they are started before the onset of acute respiratory failure [18].

IL-6 emerges as a key cytokine for the multiplication and immune evasion of both SARS-CoV-2 and cancer cells. Just to cite an example: a study found that interleukin-6 (IL-6) efficiently protects gastric cancer cells from apoptosis caused by hydrogen peroxide (H2O2) by increasing Mcl-1 production (an anti-apoptotic protein) [68].

The similarity between our proposed mechanism for IgG4-mediated immune evasion by SARS-CoV-2 and that evolved by cancer cells suggest that viruses are more complex than they are usually believed. It is not known if it were viruses or cancer cells that first developed this ingenious evasion mechanism, but we are convinced that the study of the molecular mechanisms that govern these processes could have therapeutic implications both for the treatment of viral infections and for cancer.